# Two-Dimensional Iron Phosphorus Trisulfide as a High-Capacity Cathode for Lithium Primary Battery

**DOI:** 10.3390/molecules28020537

**Published:** 2023-01-05

**Authors:** Syama Lenus, Pallavi Thakur, Sai Smruti Samantaray, Tharangattu N. Narayanan, Zhengfei Dai

**Affiliations:** 1State Key Laboratory for Mechanical Behavior of Materials, Xi’an Jiaotong University, Xi’an 710049, China; 2Tata Institute of Fundamental Research, Hyderabad 500046, India

**Keywords:** metal phosphorus trichalcogenides, discharge mechanism, cathode electrolyte interface, lithium primary battery

## Abstract

Metal phosphorus trichalcogenide (MPX_3_) materials have aroused substantial curiosity in the evolution of electrochemical storage devices due to their environment-friendliness and advantageous X-P synergic effects. The interesting intercalation properties generated due to the presence of wide van der Waals gaps along with high theoretical specific capacity pose MPX_3_ as a potential host electrode in lithium batteries. Herein, we synthesized two-dimensional iron thio-phosphate (FePS_3_) nanoflakes via a salt-template synthesis method, using low-temperature time synthesis conditions in single step. The electrochemical application of FePS_3_ has been explored through the construction of a high-capacity lithium primary battery (LPB) coin cell with FePS_3_ nanoflakes as the cathode. The galvanostatic discharge studies on the assembled LPB exhibit a high specific capacity of ~1791 mAh g^−1^ and high energy density of ~2500 Wh Kg^−1^ along with a power density of ~5226 W Kg^−1^, some of the highest reported values, indicating FePS_3_′s potential in low-cost primary batteries. A mechanistic insight into the observed three-staged discharge mechanism of the FePS_3_-based primary cell resulting in the high capacity is provided, and the findings are supported via post-mortem analyses at the electrode scale, using both electrochemical- as well as photoelectron spectroscopy-based studies.

## 1. Introduction

With an energy density of nearly 500 Wh Kg^−1^, primary batteries practically possess higher energy density and have greater long-run capability than secondary batteries, which possess low gravimetric energy density values of ~150–200 Wh Kg^−1^ [1]. In light of the plethora of applications spanning from disposable electronics to the mining industry, defense forces, aerospace technology, and medical fields, primary batteries represent a major part of the battery industry, similar to secondary (mostly Li-ion) batteries [2,3,4]. Among the lithium primary batteries, the fluorinated graphene (Li/CF_x_)-based batteries with a theoretical capacity of 865 mAh g^−1^ are well-studied, commercially available, and are the most popularly reported systems [5,6]. However, a few reports also emphasize that fluorocarbons, in which the carbon is bound to fluorine, are potent greenhouse gases on degradation (recycling), and some form toxic compounds that can accumulate in the environment [7]. Furthermore, neutralizing the fluorocarbons requires a process that includes high temperature, which drives up its cost and thus limits its adoption [7]. Additionally, CF_x_-based primary batteries are accompanied by intrinsic disadvantages that include low discharge voltage platform, inadequate rate performance, and voltage hysteresis accompanied by a severe exothermic phenomenon [8,9,10]. Researchers have also explored the other possible lithium primary batteries; however, most among those are extremely toxic and corrosive and pose harsh safety concerns [11]. Other materials for LPBs mainly include copper vanadate (CuV_2_O_6_) [12], silver vanadium oxide (Ag_2_V_4_O_11_) [13], aluminum fluoride (Li/AlF_3_) [2], and manganese dioxide (Li/MnO_2_) [14] with layered structures. However, unfortunately, the multistep reductions that occur during the insertion/intercalation of lithium in these materials results in their low specific capacity [15]. Two-dimensional metal chalcogenides (2DMCs) with carbon are a well-studied system for lithium batteries [16]. Moreover, the need for proficient energy consumption makes eco-friendly, non-carbonaceous materials more attractive and demanding.

Interestingly, when investigating the long history of 2DMCs, it was found that layered metal phosphorus trichalcogenides MPX_3_ (where M = transition metals, X = S, Se) have magnificent characteristics including exceptional intercalation characteristics and tunable bandgaps for electrochemical storage applications [17,18]. Indeed, as an anode material in LIB, MPS_3_ has a high value of theoretical capacity (>1000 mAh g^−1^), with the Li binding energy value being greater than ~1.7 eV, thus eliminating the possibility of dendrite growth. For instance, Datta et al. [19] reported the lithium ion binding energies on all the TMPS_3_ monolayers as −2.03, −2.31, −1.71, and −2.29 eV for the transition metals, TM = Mn, Fe, Co, and Ni, respectively, specifying the low possibility of Li clustering. Among all these, FePS_3_ can be a highly efficient candidate due to the presence of the most abundant TM, i.e., iron (Fe), as the constituent. Moreover, it has been explored as an electrode for lithium metal batteries at low current values [20]. In many applications, such as pacemakers, toys, devices used in military/rescue missions, etc., the use of a secondary Li battery, given their shorter discharge time, is non-recommended. Thus, it is of high significance to pursue new environmentally benign electrode materials for LPBs that could provide high energy density at lower cost. As mentioned, FePS_3_ is an ideal candidate from the TMPX_3_ family, which holds an octahedral structure that could primarily incorporate Li ions following a two-electron transfer during the reduction of Fe^2+^ to Fe^0^ [21,22]. However, research on FePS_3_ as an electrode material in LPB is still in its infant stage, predominantly due to its harsh and long-term solid-reaction synthesis protocol [23,24]. Hence, we made efforts in this direction by focusing on the development of a facile synthesis method for 2D FePS_3_ and understanding its properties for its application in a LPB at high current density.

Herein, we report the use of iron thiophosphate (FePS_3_), synthesized via a salt-template method, as an efficient lithium primary battery electrode that can deliver a high specific capacity (1791 mAh g^−1^) and energy density (2500 Wh Kg^−1^). An insight into the discharge mechanism is put forward through ante-/post-battery discharge results. Differing from its reversible behavior within the range of 1 V vs. Li/Li^+^, FePS_3_ demonstrates an irreversible discharge behavior beyond 1 V. In this work, we methodically scrutinize the capacity enhancement of the system at different voltages using various electrochemical and spectroscopic techniques. The discharge mechanism is explained as a three-stage mechanism. It is revealed here, for the first time, that the presence of defect centers and the formation of a CEI (cathode–electrolyte interface) layer eventually led to the high capacity exhibited during the first and third stages of the discharge process, respectively, while the second stage can be attributed to the lithiation in the FePS_3_ layers. Even at the high discharge current density value of 1 A g^−1^, the FePS_3_-based LPB delivers a specific capacity of 867 mAh g^−1^, which is still higher than the other reported systems, such as CF_x_ and FePS_3_ (previous reports), which were discharged at lower current density values. This study unravels the possibility of achieving high-capacity, affordable, and environment-friendly electrodes-based next-generation LPBs using FePS_3_ cathode materials.

## 2. Results

### 2.1. Synthesis of FePS_3_ Nanoflakes

Salt-template synthesis [24] can be considered as a nearly ideal synthesis strategy to prepare the compounds of MPX_3_ family because of its simplicity, low cost, low temperature, ambient growth conditions, and less time required to obtain well-grown crystals compared to the time taken with other existing synthesis methodologies. Therefore, a similar approach was adapted to synthesize the two-dimensional layered FePS_3_ using sodium chloride (NaCl) as a soft template, as shown in the Figure 1a (details are mentioned in Appendix A). The formation of FePS_3_ on the surface of the NaCl template takes place in a nitrogen atmosphere within a duration of 16 h. During the initial stages of synthesis, i.e., at 310 °C, the formation of FeS_x_ and P_x_S_y_ moieties takes place, as confirmed by X-ray photoelectron spectroscopic (XPS) studies given in the ESI. Further, yet another XPS study was conducted at the initial stage of synthesis of FePS_3_ at 310 °C (Appendix A), where a high amount of S, FeS_x_, and P_x_S_y_ were observed, and peaks corresponding to FePS_3_ were absent. Additional information with XPS spectra (Appendix A) is provided in the Appendix A. Eventually, when the temperature further rises to 500 °C, these FeS_x_ and PS_y_ moieties react to form the final product FePS_3_. The transformation can also be realized through the color change, where the dark brown mixture of FeCl_3_-coated NaCl well-ground with sulfur and phosphorus powder turns into a black-colored FePS_3_ powder after the furnace treatment (shown in ESI, Appendix A).

### 2.2. Characterisation of FePS_3_ Nanoflakes

The surface morphology of the FePS_3_ nanoflakes was analyzed by field-emission scanning electron microscopy (FE-SEM). The FE-SEM images of the samples at the different stages of synthesis are shown in Figure 1b–d. The NaCl template cubes (Figure 1b) are conformally coated with the mixture of FeCl_3_, sulfur, and phosphorus (Figure 1c). The morphology of FePS_3_ disclosed a dense cluster of nanoflakes having an average thickness of 35 nm and average areal dimensions of 350 nm, respectively. Indeed, the growth of nanoflakes on the NaCl template occurred in a stacked manner, wherein more than five nanoflakes grow on top of the FePS_3_ nanoflakes, as can be seen in the SEM images (Appendix A). Further, the SEM images of FePS_3_ nanoflakes at different magnifications are given in Appendix A. As shown in the SEM images (Appendix A), when large-sized (>10 µm) NaCl crystals were used as the templates, we obtained nanosheets of FePS_3_ compared to the nanoflakes obtained when the smaller-sized (~1 µm) NaCl template was used. The nanoflakes could provide more active edges and surface area for lithium ion storage compared to the nanosheets; hence, they were chosen for the study. SEM-assisted energy dispersive X-ray spectroscopy (EDS) analysis was also performed on the as-synthesized FePS_3_ nanoflakes. The elemental mapping confirms the presence of Fe, P, and S throughout the sample, with the relative atomic percentage being 1:1:3, as that in FePS_3_ (Appendix A). The high-resolution transmission electron microscopy (HR-TEM) analysis of the FePS_3_ sample confirms the flake-like morphology (Figure 1e). The interplanar spacing (d-spacing) of 0.29 and 0.17 nm correspond to (130) and (33$\bar{1}$) crystal planes of FePS_3_ (Figure 1f), respectively, which is further confirmed by the selected area electron diffraction (SAED) pattern (Figure 1g).

To confirm the crystal structure of the FePS_3_ nanoflakes, the X-ray diffraction (XRD) pattern of the powder sample was analyzed (Figure 2a). The diffraction pattern consists of sharp peaks at 14°, 31°, 36°, and 54° that correspond to the (001), (130), (20$\bar{2}$), and (33$\bar{1}$) crystalline planes of FePS_3_ with a monoclinic unit cell (ICSD No. 01-078-0496), which is accordance with the previous studies [25]. The miller indices (hkl) also match those obtained from the SAED pattern. Based on this information, the crystal structure (a representative structure without scaling) of the molecule is shown in the Figure 2a inset. The clear octahedral voids seen in the image would be the ideal free space for Li intercalation. The Raman spectra of FePS_3_ (Figure 2b) exhibit all the characteristics peaks of FePS_3_, including three A1g and three E_g_ modes along with one Eu mode; these are the commonly observed peaks in the Raman spectra of MPS_3_ materials [24]. The A1g and Eg peaks correspond to the in-plane and out-of-plane vibrations of the S_3_P-P_3_S unit. Another fingerprint characterization is its magnetic properties, and they are analyzed using vibrating sample magnetometry (VSM) studies (Figure 2c). The details of the measurements are given in ESI Section S4. The study confirmed the Neel temperature of FePS_3_ as 123 K (the details of the field cooling (FC) and zero field cooling (ZFC) measurements are given in the ESI), in agreement with previous studies [26]. Additionally, the thermogravimetric analysis (TGA) shows the thermal decomposition temperature of FePS_3_ at around 550 °C, in line with the reports [27].

The XPS survey spectrum of FePS_3_ nanoflakes (Appendix A) also confirms the presence of Fe, P, and S in the synthesized FePS_3_ sample, with no traces of NaCl left in the sample. The high-resolution core-level XPS spectra of Fe 2p (3/2), P 2p(3/2), and S 2p(3/2) are obtained at 712.02 eV, 132.8 eV, and 163.0 eV, respectively [28]. The atomic ratio of Fe:P:S calculated from the XPS spectrum was found to be 1:1:3, which is in agreement with the EDS findings and hence ensures the stoichiometry FePS_3_. The layered FePS_3_ possess the same monoclinic structure as that of CdX_2_ (X = Cl, I) where the metallic centers are replaced by Fe^2+^ and P-P (P_2_) pairs. Consequently, the overall geometrical arrangement relies on two distinct octahedral units, FeS_6_ and P_2_S_6_. Further structural details can be easily understood by considering the doubled formula unit, i.e., Fe_2_P_2_S_6_ with one P_2_ pair. Each S atom is bonded with one P and two Fe^2+^ centers [29]. The P_2_ pair possesses a formal valency of +8 (P_2_^+8^), where each P contributes 3s2 and 3p2 electrons in the valence-band states and one remaining electron for P-P bond formation. In addition to this, each S and Fe contributes 3s2, 3p4, and 4s2 outer-shell electrons, respectively. Therefore, in a Fe_2_P_2_S_6_ unit, a total of 48 electrons fill up the valence-band states, and the Fe^2+^ metal cores are stabilized by the [PS_3_]^−2^ units [29].

### 2.3. Cathode Properties of FePS_3_ Material

The electrochemical discharge performances of the FePS_3_ system as a cathode material in an LPB is checked using a Li ion half cell, the details of which are given in the experimental details section. The electrochemical performance of the as-synthesized FePS_3_ coated on copper foil was investigated using CR2016 coin cells. CR2016 coin cells were assembled using the conventional method. Coin cell testing specifications used in sample preparation for the characterizations are given in Appendix A. The cell showed an open-circuit voltage (OCV) of 3.2 V when assembled and was galvanostatically (10 mA g^−1^) discharged until 0.5 V vs. Li/Li^+^ (foil) as the anode. Figure 2d shows the schematic representation of a lithium primary coin-cell assembly with FePS_3_ nanoflakes as the cathode material. The cyclic voltametric (CV) test was performed at a scan rate of 0.1 mV s^−1^ to analyze the electrochemical redox reactions involved. The two reduction peaks (Figure 2e) can be found near 1.4 and 1.1 V, indicating the two stages of lithiation in FePS_3_. Similarly, during the anodic sweep, the oxidation peaks located at 1.9 and 2.2 V vs. Li/Li^+^ can be correlated to the corresponding de-lithiation steps. The cathodic current in the region from 0.9 V to 0.5 V could be ascribed to the irreversible reaction between the active materials and the electrolyte towards the formation of the CEI (cathode electrolyte interface), which is discussed later. To confirm the lithium storage mechanism in the FePS_3_ electrode, a freshly assembled cell was galvanostatically discharged at a current density of 10 mA g^−1^, as shown in Figure 2f. The voltage plateaus located at around 1.6 V and 1.25 V in the discharge profile are in accordance with the CV data. The discharge capacity with respect to the total mass loading of the active material FePS_3_ is found to be 1791 mAh g^−1^, which is about 3.5 times that of fluorographene (520 mAh g^−1^), which is one of the more common primary battery systems. Additionally, the galvanostatic discharge performance and CV data of FePS_3_ as a secondary battery are shown in Appendix A.

### 2.4. Post-mortem Analyses of FePS_3_ Material

We emphasize that the obtained discharge capacity of 1791 mA h g^−1^ is significantly greater than that of the theoretical value for FePS_3_ (1318 mAh g^−1^) [22]. From the discharge plateau (Figure 2f), it is suspected that the total capacity contribution could be due to three distinct reactions that occur in the three different stages. The first stage could be due to the presence of unreacted defect moieties present in the sample. The second stage, which is the main contributor to the capacity, could be the active material FePS_3_, and the sloping voltage plateaus that occur below 1 V could be the third stage, which is the formation of a stable CEI film on the surface of the electrode (in accordance with the CV plot).

In order to verify our proposition, the post-mortem analysis of the battery cells discharged at different stages was conducted using XPS- and SEM-based analyses. The chemical state information of the pristine and discharged electrodes, acquired through XPS, give significant understanding of the mechanism. The detailed mechanism of all the three stages is discussed below. As shown in Figure 3, the XPS investigation of the ante- and post-discharged cells at the photoemission lines of S 2p, P 2p, Fe 2p, and Li 1s (Appendix A) were recorded. From now on, the uncycled electrode prior to discharge is named as pristine FePS_3_, and the cells discharged until 1.5 V, 1 V, and 0.5 V are named as DChg_1.5 V, DChg_1 V, and DChg_<1 V.

From Figure 2f, it can be identified that the second stage is the main contributor towards the capacity, i.e., due to the main component FePS_3_. The electrochemical discharge reaction mechanisms of Li ion insertion in FePS_3_ were studied earlier and can be described as in Equations (1)–(3):
(1)$$2\mathrm{Fe}PS_{3}+3Li^{+}+3e^{-}\;↔\;3Li_{1.5}\mathrm{Fe}PS_{3}$$
(2)$$Li_{1.5}\mathrm{Fe}PS_{3}+Li^{+}+e^{+}\;↔\;2Li_{2}\mathrm{Fe}PS_{3}$$
(3)$$Li_{x}\mathrm{Fe}PS_{3}+\left(6-x\right)Li^{+}+\left(6-x\right)e^{-}\;\to 3Li_{2}S+\mathrm{Fe}+P\left(x=1.5\;\mathrm{or}\;2\right)$$

Initially, at x = 1.5, topochemical storage mechanism through geometric factors, i.e., filling of octahedral voids, would contribute up to 219 mAh g^−1^ to the capacity (Equation (1)). Thereafter, the formation of Li_1.5_ FePS_3_, and the further addition of Li ions can proceed through Equations (2) and (3) simultaneously. For x = 2, the d-levels of Fe^2+^ cations act as acceptor centers, where Li ion intercalation is accompanied by a reduction of cations (Fe^2+^ to Fe^0^) in the MPX_3_ structure. This conversion shows a discharge capacity of 292 mAh g^−1^ following Equation (2). Nevertheless, the higher discharge capacity of 1318 mAh g^−1^ is achieved due to the irreversible changes that occur in the active cathodic material, resulting in Li_2_S as the final discharge product, as shown in Equation (3). To confirm that the above intercalation is followed by a conversion mechanism, all the discharged electrodes (DChg_1.5 V, DChg_1 V, and DChg_<1 V) were investigated through XPS.

The high-resolution Fe 2p XPS spectrum of Dchg_1.5 V sample shows two spin-orbit-coupled peaks at 711.1 eV (Fe 2p3/2) and 724.8 eV (Fe 2p1/2) (Figure 3d). It can be noted that the Fe 2p3/2 binding energy (BE) peak is downshifted by 0.9 eV compared to the pristine FePS_3_ peak at 712.0 eV (Figure 3a), indicating the formation of Li_x_FePS_3_ (x ≤ 1.5), accompanied by the reduction of Fe^2+^ in accordance to Equation (1). The further downshift of Fe 2p3/2 binding energy for the sample Dchg_1 V (Figure 3g) indicates an increase in the Li content (x > 1.5) in Li_x_FePS_3_ as in Equation (2), possibly by 2e^−^ reduction of Fe^2+^ to Fe0. Moreover, the decrease in the peak intensity of Fe 2p3/2 for Dchg_1 V indicates the increase in the irreversible reactions as in Equation (3). Similarly, the comparison of the high-resolution P 2p (2p 3/2 and 2p 1/2) XPS spectrum of the pristine FePS_3_, Dchg_1.5 V, and Dchg_1 V samples also supports the above-mentioned hypothesis. The slight consecutive upshifts in the binding energy of 2p 3/2 from pristine FePS_3_ (132.9 eV) to Dchg_1.5 V (133 eV) and further to Dchg_1.5 V (133.1eV) are visible in Figure 3b,e,h. These indicate the replacement of Fe centers by more electropositive Li in -P-S-Fe- bonds during Li-ion insertion at the Fe^2+^ centers, followed by their reduction and formation of Li_x_FePS_3_ (x ≥ 1.5). The high-resolution S 2p XPS spectra for all three samples do not show any appreciable change for the binding energy peak at around 162.9 eV. However, it is taken into consideration for determining the structural dissociation mentioned in Equation (3). The enhancement in the intensity of the binding energy peak at around 161.6 eV corresponding to Li_2_S relative to that of the Li_x_FePS_3_ from Dchg_1.5 V to Dchg_1 V confirms the overall collapse of the Li_x_FePS_3_ structure upon further addition of Li ions and formation of Li_2_S as the final discharge product.

The lithium intercalation and conversion mechanism discussed above during the discharge in FePS_3_ are further supported by high-resolution Li 1s spectra. The Dchg_1.5 V sample shows the presence of both Li_x_FePS_3_ and Li_2_S,as the Li 1s binding energy peaks for both are overlapping at 55.6 eV. However, as the discharge depth increases in Dchg_1 V, a slight downshift of 0.2 eV is observed in the binding energy peak that can be ascribed to the increase in Li_2_S concentration, as shown in Equation (3).

Hence, the main lithium storage mechanism for FePS_3_ (second stage) is confirmed to proceed by the formation of the Li_x_FePS_3_ initially (x ≥ 1.5), followed by the formation of Li_2_S on the subsequent addition of Li ions (Equation (3)), which gives rise to a discharge capacity of 1318 mAh g^−1^. However, the observed discharge capacity of 1791 mAh g^−1^ can be explained as the contribution of two different stages (Figure 2f): the first stage and third stage (discussed later). Before that, a question that may arise is whether the Fe or P that are formed in accordance to Equation (3) act as extra storage sites for lithium. We can clearly omit this possibility, as no peak corresponding to either Li_3_P or any lithium–iron compound formation is observed in the high-resolution Li 1s, P 2p, and Fe 2p XPS spectra. Instead, the spectra show the presence of Fe-F and P-F bond formation with the increase in the depth of discharge, as shown in Figure 3d–h. This can be attributed to the iron-catalyzed dissociation of the electrolyte, which is LiPF_6_, at a longer discharge period. Such dissociations are also reported for different transition metals [30,31]. For instance, J Park et al. [32] reported that Fe-based electrodes promote severe electrolyte decomposition (especially PF_6_-based electrolyte) and subsequent growth of a thick interface layer. In this case also, on dissociation of LiPF_6_, the P-F bond formation happens prior to any lithium intercalation in P (discussed in detail in the later part).

## 3. Discussion

### 3.1. Reasons for the Additional Capacity

The observed extra storage capacity in comparison to the theoretical limit is due to the contribution from the other two stages, as discussed here.

As explained in the beginning (first stage), the formation of FePS_3_ happens via the formation of the FeS_x_ and PS_y_ units. However, some of the unreacted sulfur (S8) along with defect moieties generated during the synthesis, such as FeS_x_ and P_x_S_y_, contribute to the extra capacity. Their presence along with the pristine sample is confirmed by the high-resolution XPS spectra of pristine FePS_3_. In high-resolution S 2p spectra, the peaks at binding energy 164.4 eV and 161.6 eV correspond to the presence of sulfur (S8) and FeS_x_, respectively, while the S 2p binding energy peaks for P-S-P moieties are supposed to overlap with FePS_3_ at around 162.9 eV. Additionally, the high-resolution Fe 2p signals at binding energy 716.30 eV (Fe 2p 3/2) and 729.6 eV (Fe 2p 1/2) correspond to the presence of FeSx. On the other hand, the sharp peak at 131.2 eV in P 2p high-resolution spectra indicates the presence of P-S-P moieties. Therefore, the first stage depicts the lithium storage in these three moieties according to the following equations:
(4)$$16Li^{+}+S_{8}+16e^{-}\to 8Li_{2}S1$$
(5)$$FeS_{x}+8Li^{+}+8e^{-}\to 4Li_{2}S+FeS_{x-4}$$
(6)$$PS_{y}+xLi^{+}+xe^{-}\;\to \;Li_{x}PS_{y}$$

The high-resolution XPS spectra of the discharged electrodes, discussed before, was further analyzed to confirm the discharge reactions mentioned in the Equations (4)–(6) during the first stage. Unlike the second stage, where Li_2_S was the final discharge product after the FePS_3_ dissociation, Li_2_S is formed as the result of unreacted S and FeS_x_ moieties present in the sample in the first stage. The total contribution of such defect moieties towards the capacity is particularly smaller (<250 mAh g^−1^). It is important to note that the sulfur-based cathodes are limited by the low practically achievable energy density. For instance, in 2015, Y. Ma [33] and colleagues reported a Li-S battery that could achieve an energy density of hardly 504 Wh Kg^−1^ even though the theoretical energy density was 2600 Wh Kg^−1^.

As discussed in the second stage, the high-resolution XPS spectra of S 2p and Li 1s (ESI) for Dchg_1.5 V shows the presence of both Li_x_FePS_3_ and Li_2_S. The Li_2_S present here is attributed to the discharged product formed in the first stage due to the three other moieties present in the system (Equations (4) and (5)), while the increase in the Li_2_S content in Dchg_1 V is due to the disintegration of the Li_x_FePS_3_ with the further addition of Li ions. The other contributor towards the additional capacity is the P_x_S_y_ moiety, according to Equation (6). The binding energy for Li_x_PS_y_ in high-resolution P 2p and S 2p XPS spectra overlays with that of the other discharge products, Li_x_FePS_3_ and Li_2_S, respectively (Figure 3e,f,h,i). As discussed above, the second stage corresponds to the interaction of lithium in to FePS_3_.

In the third stage, the extra capacity can be assigned to the Li ions reacting with the decomposed products of the electrolyte and resulting in the formation of a CEI. At high depths of discharge, the samples Dchg_1 V and Dchg_<1 V show the emergence of LiF along with the main discharge products. The presence of the peak at binding energy of 56.2 eV in the high-resolution Li 1s spectra of Dchg_1 V and Dchg_<1 V clearly shows the dominance of the LiF phase (Appendix A). As mentioned earlier, in the second stage, the Fe-based electrodes or metallic iron formed during the discharge step of Equation (3) can catalyze the PF6-based electrolyte decomposition, as shown in Equation (7), followed by the LiF formation in Equation (8):
(7)$$PF_{6}\;\to \;PF_{5}+•F$$
(8)$$Li^{+}+•F\;\to \;LiF$$

LiF is the main component of the CEI and contributes towards the extra capacity. Such electrolyte (salt) decomposition resulting in the formation of LiF is in accordance with the previous reports where the CEI only contains salt-based products such as LiF, Li_x_F_y_, and Li_x_PO_y_F_z_ [34]. LiF has a high theoretical specific capacity of around 1000 mAh g^−1^ and hence contributes to the extra capacity towards the end. Even though the system discussed here is a primary battery, the study of the CEI formation can be extended to the secondary systems as well. A stable CEI is imperative in order to increase the thermal stability of the aggressive cathodes, evade the inter-face side reactions, and guarantee the electrochemical performance. CEI with high inorganic content (for example, Li_2_CO_3_ and LiF) would function as an electronic insulator and facilitate the desired Li+ transport. A well-regulated growth of robust cathode–electrolyte interphase (CEI) with high inorganic content is the most beneficial approach to tackle the thermal runaway concerns [35].

The above-mentioned three-stage mechanism is further supported by SEM (Figure 4a–d). The consistent increase in the cross-sectional thickness from the pristine FePS_3_ to Dchg_1.5 V and finally to Dchg_1 V electrode confirms the lithium intercalation over the increased discharge depth (supporting information, Appendix A). The elemental mapping is also performed by EDS of the fully discharged sample. The Figure 4a–d show the presence of sulfur (S) (atomic percentage = 5%) until a certain thickness, indicating the presence of Li_2_S as in Equation (3) during the initial stage of discharge. The percentage of other elements starts increasing while moving towards the top, i.e., Fe (atomic percentage = 2.4%) and F (atomic percentage = 25%), indicating the formation of fluorine-based decomposition products (discussed later). The high percentage of fluorine relative to other elements present indicates the presence of LiF as the main product towards the end of the discharge (Equation (8)), in tune with the XPS measurements. Hence, the main contributor towards the total capacity is the FePS_3_ (discussed in second stage), while the defect moieties and CEI formation result in the extra capacity, as discussed in first and third stage, respectively. We expect that such controlled formation of CEI layer via LiF formation would open up innovative ways of engineering CEI formation strategies, as it is crucial for the performance of various secondary batteries as well [36].

### 3.2. Electrolyte Decomposition

In general, primary batteries are known to run for longer periods. Hence, both during the discharge and after the complete utilization of a battery, the study of the degradation products is important in two aspects. Firstly, treatment of the used components of the battery becomes easier with prior knowledge of the species present in the system [37,38,39]. Secondly, in biological systems where the batteries are used for longer runs, the formation of the hazardous chemicals is undesirable [40,41]. In this study, the main changes (electrolyte decomposition) are also observed at lower discharge voltages. These are the reactions that caused the further formation of CEI products in the Dchg_<1 V electrodes, confirmed by high-resolution Fe 2p, P 2p, S 2p, (Appendix A), C 1s (Appendix A), and Li 1s XPS spectra.
(9)$$2LiF+Fe\to xLi^{+}+\mathrm{Fe}+iFeF_{2}$$
(10)$${\mathrm{Fe}}^{*}+\mathrm{EC}+F\to FeOCOOCH_{2}F$$
(11)$$\mathrm{yF}+xROCOOR\to {\left(\mathrm{RO}\right)}_{x}P\left(O\right)F_{y}$$
(12)$$Li_{2}S+ROCOOR\to LiSO_{3}+\mathrm{dissociation}\;\mathrm{products}$$

The uniform distribution of phosphorus, oxygen, and carbon on the surface of the discharged electrodes also supports the electrolyte decomposition pathways, as shown in Equations (9)–(12) and given in electronic Appendix A.

### 3.3. Three-Stage Discharge Mechanism in FePS_3_ Systems

Based on the findings acquired through XPS and SEM analyses, we propose a three-stage discharge mechanism for the FePS_3_ system, validating the obtained very high capacity (1791 mAh g^−1^) (Figure 2f). During the first stage, defect species that developed on account of unreacted S, FeS_x_, and P_x_S_y_ are recognized in the sample that contribute towards the initial capacity. However, the overall capacity contribution from such defect moieties is explicitly less than 250 mAh g^−1^. The main discharge product during this stage is Li_2_S due to the lithiation in defect moieties (first stage, shaded in blue color). In the second stage, the FePS_3_ itself is the principal capacity storage contributor. Meanwhile, the discharge proceeds via lithium intercalation and conversion mechanism within FePS_3_ to form Li_x_FePS_3_ (x ≤ 2) along with the formation of dissociation products including Li_2_S, P, and metallic Fe (second stage, shaded in pink color). At this stage, the initial formation of CEI is also recognized. The third stage includes the decomposition of the electrolyte that captures the Li ions, causing a stable CEI formation (particularly LiF) and contributing towards the excess capacity (third stage, shaded in yellow color). Moreover, several irreversible reactions are also occurring due to electrode/electrolyte decomposition, as shown above.

### 3.4. Performance of FePS_3_ in LPB

The primary cells should function at high discharge loads as well as work for a vast range of applications. Hence, we studied the rate capability at different current densities for the FePS_3_-assisted Li ion cells. The cells were discharged at the current densities of 10, 50, 100, 250, 500, and 1000 mA g^−1^. The plots are shown in Figure 5a. It was observed that even at the higher current densities of 500–1000 mAg^−1^, the cells show high discharge capacities around 1000 and 900 mAh g^−1^ (Figure 5b). This shows the high efficiency of LiFePS_3_ as a primary storage material even for high loads. Further, a full cell experiment using lithium as cathode instead of lithiated graphite delivered a high capacity of 1131 mAh g^−1^, presenting its exceptional behavior as a primary battery. More details of the full cell experiment are given in Section S11 ESI. A comparison plot of FePS_3_ battery performance with other reported materials is shown in Figure 5c and Appendix A. The high capacity and high energy density of the FePS_3_ showcase the best value among previous reports. Additionally, a table is tabulated (Appendix A) to give a compact view of the reported primary batteries with specific capacity, energy density, current density, and voltage window. Finally, a small motor (1.2 V/15 mA) was operated (video of the working motor is shown in the supporting information, Video S1) using the assembled battery for demonstration purposes (Figure 5c inset) to indicate the high capacity of the FePS_3_-based primary cells. The configuration for the demonstration was adapted from a previous study [42]. The movie Appendix A displays the motor running for around 45 s using a single coin cell. The Figure 5c inset shows the on and off state of the motor. Furthermore, the shelf life of the battery was calculated using an aged cell kept for one month after assembling. The aged cell exhibited specific capacity of 1771 mAh g^−1^ after one month, which is almost nearer to a freshly assembled cell (Figure 5d). The overlapping, overlaid data of fresh and aged cells show no evidence of any cathodic corrosion, electrolyte degradation, or side reactions, highlighting the superiority of FePS_3_ as a lithium primary battery without any self-discharge. The cost analysis of a single coin cell was estimated using the details given in Appendix A. The estimated cost is USD 0.17, in which USD 0.15 is the cost of assembling parts. The given cost analysis indicates the possibilities of low-cost battery development using FePS_3_ as the cathode material.

## 4. Materials and Methods

### 4.1. Experimental Section

Materials: Iron (III) chloride hexahydrate (FeCl_3_·6H_2_O), sulfur powder, and lithium hexafluorophosphate (LiPF_6_) in DMC solvent were purchased from Sigma Aldrich Inc. St. Louis, MO, USA. Phosphorus red was purchased from Sisco Research Laboratories Pvt Ltd, Mumbai, India. Sodium chloride (NaCl) extra pure was purchased from S.D fine Chem Limited (Chennai, India). 1-Methyl-2 pyrrolidinone (NMP) was purchased from Avra Synthesis Pvt Ltd, Hyderabad, India.

### 4.2. Preparation of FePS_3_ Electrode

The electrode preparation was performed in the traditional way by grinding 80% of FePS_3_ material, 10% carbon black, and 10% (polyvinylidene fluoride) PVDF into a semi-thick slurry using NMP. The slurry was coated on battery-grade copper foil using a doctor blade. The electrode was dried at 80 °C for 8 h. Circular discs (diameter = 1cm) were cut off from the same and used to assemble the CR2016-coin cell battery with lithium foil as the counter electrode. The typical mass load of active material was controlled to be about 1 to 2 mg cm^−2^.

### 4.3. Characterization Techniques

To determine the vibrational Raman modes of FePS_3_, Renishaw inVia Raman microscope (532 nm excitation) was used. Further, to know the surface topography and composition of the sample, a field emission scanning electron microscope (FESEM) JEOL JSM-7200F (JEOL Ltd, Tokyo, Japan) along with EDS was used. A transmission electron microscope TEM, JEM-2100F (JEOL Ltd, Tokyo, Japan) with acceleration voltage of 300 kV (aberration corrected) was used for TEM analysis of the samples. The crystallinity of the materials was studied using X-ray diffraction (Bruker Tensor-27). X-ray photoelectron spectroscopy (XPS) results from Thermo Fisher Scientific (Cleveland, OH, USA) ESCALAB Xi^+^ provided the surface elemental composition and electronic states of the samples. Thermogravimetric analyses (TGA) were operated with a TA Instruments (TGA 550, TA instruments, New Castle DE, USA) apparatus at a heating rate of 10 °C min^−1^ under nitrogen flow. 

### 4.4. Electrochemical Measurements

A Biologic SP-300 potentiostat (Biologic, Vaucanson, France) was used for the electrochemical measurements (CV). The cyclic voltametric tests were carried out using an adapter with a two-electrode cell, which was assembled using the FePS_3_ electrode as the working electrode and lithium metal as both the counter and reference electrodes. The CV was performed with a 0.1 mV/s scan rate. Further, the battery cell assembly was carried out in an Ar-filled glovebox with <0.1 ppm H_2_O content. Moreover, galvanostatic charge and discharge tests were conducted using a Neware battery-testing system in the voltage range of 0.01 to 3.0 V vs. Li/Li^+^.

## 5. Conclusions

A layered metal phosphide trichalcogenide-based, i.e., FePS_3_-based, economically viable, environmentally benign, high-capacity cathode material for LPB is demonstrated herein. The salt-template-assisted synthesis method provided here put forward a scalable but simple and efficient synthesis route for FePS_3_ nanoflakes. Moreover, the Li ion cells constructed showed a very high rate capability (1 A g^−1^) and specific capacity (867 mAh g^−1^) compared to that of the previously reported primary battery electrode materials. A detailed post-mortem analysis unraveled the discharge mechanism in FePS_3_, leading to its high capacity. A three-staged discharge mechanism is proposed in accordance with the obtained results. The LPB assembled with FePS_3_ show a high discharge capacity of 1791 mAh g^−1^ and a high energy density of 2500 Wh Kg^−1^, with a long shelf life. The FePS_3_ LPB delivered a maximum specific capacity of 867 mAh g^−1^ even at a high current density of 1 A g^−1^, which is higher than all the previously reported systems. This work provides a simple and low-cost approach to develop MPX_3_-based, highly efficient primary battery systems, underling its capability towards a vast range of practical applications.

## Figures and Tables

**Figure 1 molecules-28-00537-f001:**
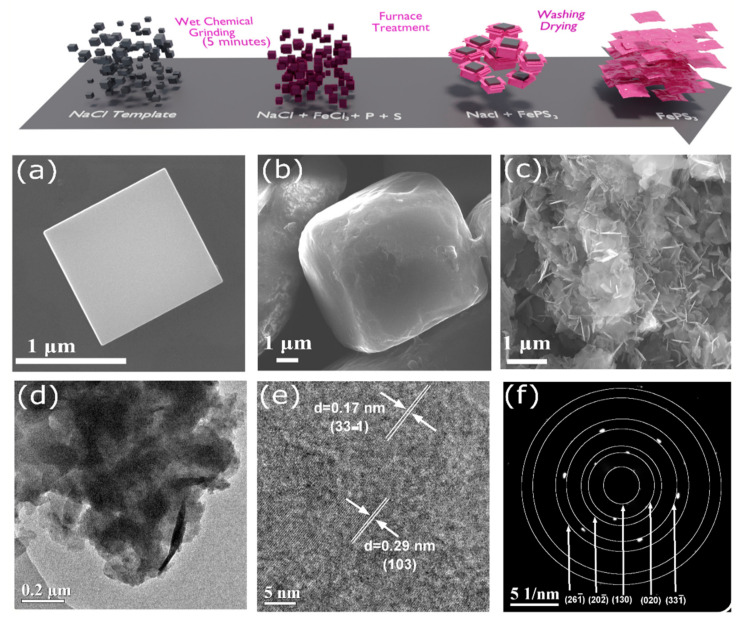
Schematization of the synthesis strategy of FePS_3_ via the salt-template method. SEM images of (**a**) NaCl template (**b**) FeCl_3_, sulfur, and phosphorus-coated NaCl and (**c**) FePS_3_ nanoflakes. (**d**) TEM image of FePS_3_ nanoflakes; (**e**) HRTEM image of a representative FePS_3_. The well-defined interlayer spacing of 0.17 and 0.29 nm indicates the presence of crystalline FePS_3_. (**f**) Selected area electron diffraction (SAED) pattern of FePS_3_ nanoflakes.

**Figure 2 molecules-28-00537-f002:**
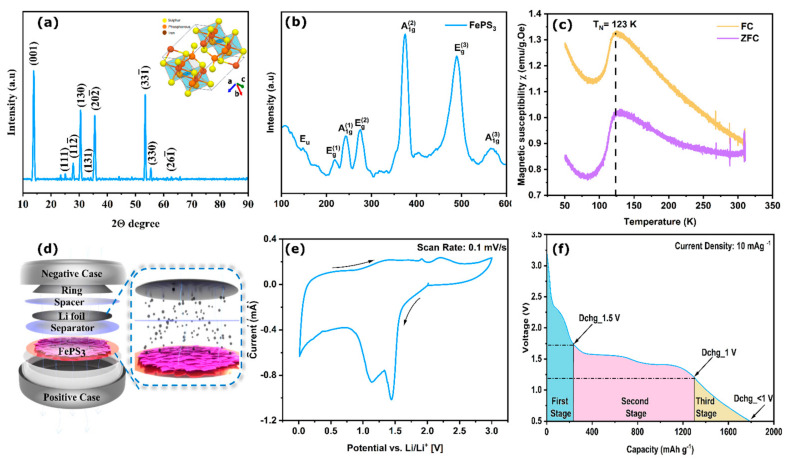
(**a**) X-ray diffraction pattern, (inset: geometric model of FePS_3_ molecule with octahedral voids), (**b**) Raman characterization, and (**c**) vibrating sample magnetometer (VSM) analysis of as-synthesized FePS_3_. χ vs. T plot for pristine FePS_3_ in the temperature range of 50–300 K under an applied magnetic field of H = 100 Oe. FC and ZCF represent field cooling and zero field cooling, represented by blue and green solid lines. (**d**) Schematic representation of lithium primary battery with FePS_3_ nanoflakes as cathode. (**e**) CV plots at 0.1 mV s^−1^. (**f**) Galvanostatic full-discharge profile of FePS_3_ cathode with proposed three-stage mechanism.

**Figure 3 molecules-28-00537-f003:**
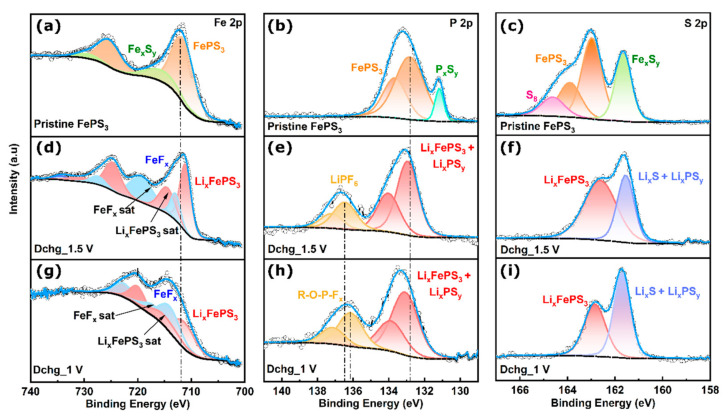
Comparison of high-resolution XPS spectra of pristine FePS_3_ and discharged FePS_3_ lithium primary battery during different stages of discharge. (**a**,**d**,**g**) High-resolution XPS spectra of Fe 2p; (**b**,**e**,**h**) high-resolution XPS spectra of P 2p; and (**c**,**f**,**i**) high-resolution XPS spectra of S 2p of pristine FePS_3_ and discharged FePS_3_ until 1.5 and 1 V.

**Figure 4 molecules-28-00537-f004:**
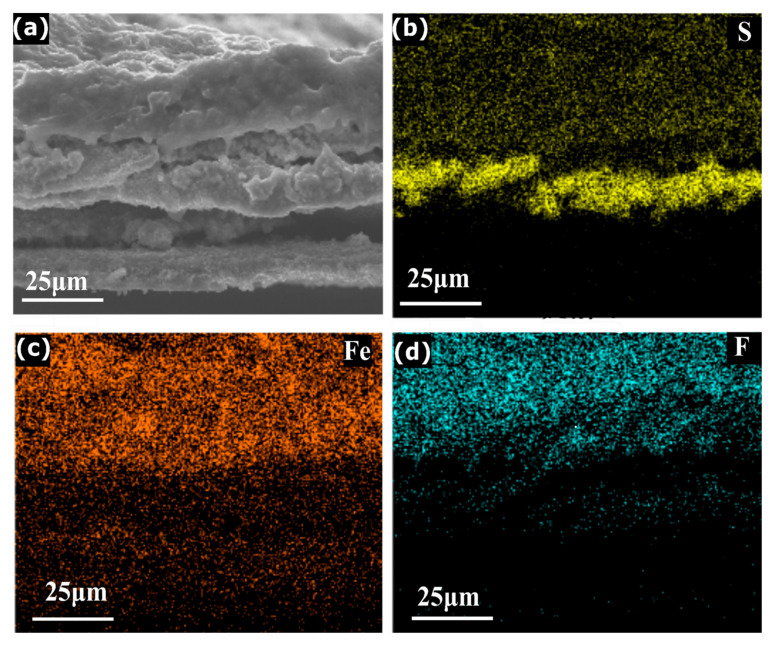
SEM and EDS mapping analysis of a cross-section of the FePS_3_ electrode discharged till 0.5 V: (**a**) secondary electron image of SEM and (**b**) S (yellow), (**c**) Fe (orange) and (**d**) F (blue) elemental mapping using EDS.

**Figure 5 molecules-28-00537-f005:**
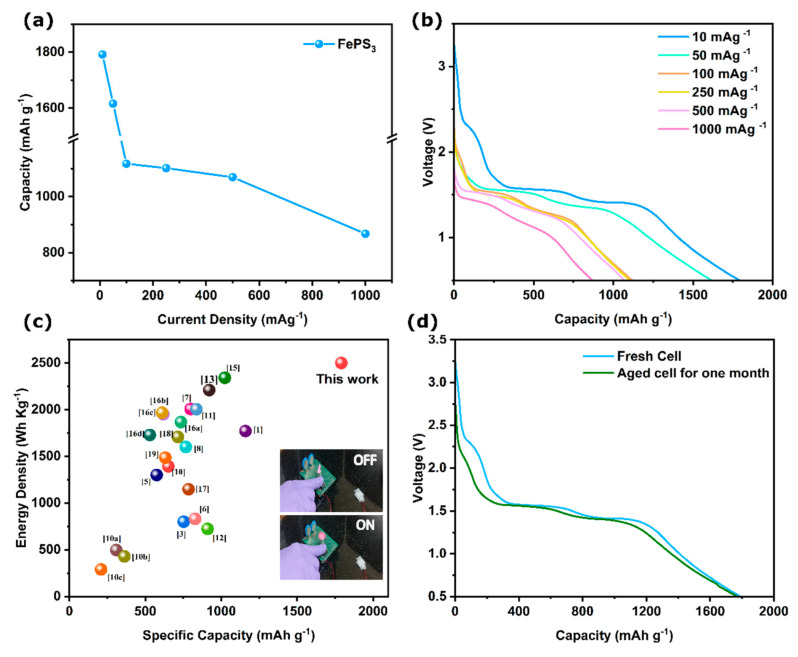
(**a**) Galvanostatic full-discharge profiles and (**b**) rate capability of FePS_3_ primary battery at various current densities. (**c**) Comparison analysis graph of reported batteries with this work with the photographic images of FePS_3_-based primary cell (battery) running a small motor (1.2 V/15 mA) as inset. (**d**) Shelf-life study of the battery tested one month after assembling, where the cells are discharged at 10 mA g^−1^.

## Data Availability

Not applicable.

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
