# Peer review of "Two-Dimensional Iron Phosphorus Trisulfide as a High-Capacity Cathode for Lithium Primary Battery"

_molecules, 2023, doi:10.3390/molecules28020537_

Round 1
Reviewer 1 Report
A FePS3 nanoflakes cathode material for Lithium primary battery is investigated. The data seems promising. However, the manuscript rises a lot of questions than answers. A comprehensive study of the mechanism is required to justify the conclusion. Thus, I don’t suggest this manuscript is ready for publication. My questions and comments are as below:
1. The TGA data doesn’t show a clear trend. It is also different from the cited paper. There is a possibility that the sample is a mixture with a certain amount of other materials than just FePS3.
2. Energy density was provided but the whole manuscript was focused on the half cell which uses lithium foil as the counter electrode. There is no description for the full cell and the anode for the full cell. And no information on how the energy density is calculated.
3. The mechanism is still unclear. Most discussion is hypothesis than characterization. For example, for the additional capacity, authors assumed that it is because of unreacted sulfur or FeSx or PxSy during the synthesis. This means that the materials are likely a mixture. So, what percentage of each component exists in the mixture? The specific capacity is calculated based on pure FePS3, which is also incorrect. Nevertheless, there is no characterization to help solve the mystery.
Thus, It is suggested that the authors will explore the mechanism comprehensively or find out possible mistakes during the synthesis and justify the conclusion in an improved manuscript.
Author Response
Please see the attachment about the author's reply to the review report.
We appreciate the valuable comments from the reviewers. Indeed, the reviewers’ advices and recommendations were very prompt and helpful. We revised the manuscript according to the reviewers’ directions. Thank you very much.

Reviewer 2 Report
Review comment.
This paper utilized a salt template method in synthesizing FePS3 and used it for the cathode of primary lithium metal batteries. The paper is systematic but is with some questionable points. I can approve on the acceptance of the paper only if the following issues are well addressed.
1. The introduction should be appropriately changed. The paper [20] has actually dealt with also primary lithium metal batteries, rather that lithium ion batteries as the authors described (page 2, line 62-63). The purpose of the paper needs to be clearly described in the introduction part.
2. How does the template size affect the synthesis process and resulting product? Have the authors tried to control this?
3. XPS related to the initial stage of synthesis at 310 oC is missing (page 3, line 101-102).
4. The sentence 105-106 on page 3 is quite misleading. FePS3 should be formed upon heat treatment not upon mixing the precursor powders.
5. The SAED did not show diffraction points for (130) and (202) planes, which were pretty strong in the XRD profile. How to explain this?
6. On page 5, line 195-196, the authors compared the capacity of FePS3 with graphite. The purpose of making such comparison is a bit confusing for readers.
7. S usually tend to evaporate when the temperature is higher than 200 oC, why they can still exist after annealing at 500 oC? Please detail the heat treatment a bit more. Was it a closed system or open system?
8. It is not appropriate to use the same information (Figure 2f) for FOUR times in the same paper content. Please consider to rearrange the data.
Author Response

(The authors gave the same response as above.)

Round 2
Reviewer 1 Report
Answered all my questions.
Reviewer 2 Report
The paper is acceptable in the current form.